# "You do need each member of the team to bring that next piece of the puzzle": Allied health professionals' experience of interprofessional complex care in hospital settings

Felice Borghmans [1,2]*, Venesser Fernandes[1,3,4], Stella Laletas[1,5,6,7], Harvey Newnham[1,2,8,9,10]

**1** Faculty of Education, Monash University, Clayton, Victoria, Australia, **2** Alfred Health, Melbourne, Victoria, Australia, **3** Australian College of Education, Melbourne, Victoria, Australia, **4** Royal Society of Arts, London, United Kingdom, **5** Fronditha Aged Care, Mulgrave, Victoria, Australia, **6** To Be Loved Network, Kew East, Victoria, Australia, **7** Pacifica Congress, Unley, South Australia, Australia, **8** Safer Care Victoria, Melbourne, Victoria, Australia, **9** Melbourne Health, Melbourne, Victoria, Australia, **10** Monash University Clinical School, Melbourne, Victoria, Australia

* felice.borghmans@monash.edu, feliceborghmans@gmail.com

## Abstract

This study explores the experiences of allied health professionals who work in interprofessional hospital complex care teams. The aim of the study was to identify factors influential to meaningful clinician experiences in these contexts. Increase in interprofessional complex care in hospital settings reflects rising population health complexity. Furthermore, growth in these models coincides with a heightened focus on health system efficiency due to rising healthcare costs, resource constraints, and health workforce shortfalls. Combined, these issues constitute a 'wicked problem'. However, research exploring the experiences of clinicians working under these conditions is limited, exposing the knowledge gap of interest to this study. Using a qualitative approach, in-depth interviews were conducted with allied health professionals engaged in hospital-based interprofessional complex care, and their narratives were analysed according to the conceptual framework of complex adaptive phenomenology. The study identified four interconnected themes: workplace culture and leadership, interprofessional practice, healthcare ethics, and the ambiguity of complex care. Furthermore, the notion of 'empowerment of self and others' was a continuous thread throughout, which appeared essential to effective interprofessional practice. The study showed how the clinician experience provides a window to the functioning of a healthcare system and the bearing of experience on healthcare efficiency and sustainability. Recommendations include developing a more balanced approach to 'efficiency' in policy settings, implementing structured leadership development programs within the allied health workforce, empowering under-graduate practitioners through education to work effectively with uncertainty, and increasing research into the clinician experience of interdisciplinary complex care practice.

**Data availability statement:** Data are available from the Institutional Data repository via the following link: https://doi.org/10.26180/24798981.v1

**Funding:** The author(s) received no specific funding for this work.

**Competing interests:** The authors have declared that no competing interests exist.

## 1. Introduction

This study explored the experiences of allied health professionals working in interprofessional hospital complex care teams. The study aimed to identify factors influential to meaningful and engaging clinician experiences in these contexts. The term "allied health" was defined as health professions that collaborate or work alongside medical and nursing professionals, such as the social worker and physiotherapists who participated in the study [1,2].

Thanks to increased appreciation for disciplinary and perspectival diversity, allied health professionals are now essential members of interprofessional hospital care teams [3]. Yet, while such teams are now ubiquitous, the experience of clinicians practicing in these contexts has attracted less attention [4]. Attention is warranted, however, as modern health systems face multiple concurrent challenges that influence the experience of providing care. These include population ageing and chronic conditions prevalence that increase demand for more complex health care [5,6]. Rising rates of burnout among clinicians and a shortage of health care workers add to these concerns [7]. Together, these issues are exacerbated by an unsustainable rise in healthcare costs, forcing healthcare decision makers to focus on cost containment and efficiency, which then impact the clinician experience of care giving [7,8]. Hence, to ensure that institutional settings can support and sustain a high calibre, motivated, and effective interprofessional workforce, it is important to understand how allied health clinicians experience complex care practice in such a dynamic context.

In this study, in-depth interviews were undertaken with three experienced allied health practitioners, each working in different care settings: a general medical ward, a subacute geriatric and rehabilitation setting, and an ambulatory rehabilitation setting. The conceptual framework of this study employed the novel approach of complex adaptive phenomenology (CAP) that was developed by two of the researchers (FB and SL) [9]. The framework offers a complexity informed perspective on the phenomenon of experience [9] and is thus well-matched to the research context.

The study is presented as follows. Section two provides a brief account of interprofessional practice in hospital settings. Section three explains the conceptual framework for this research, in terms of how complexity theory and phenomenology come together in CAP to generate an interconnected and dynamic account of clinician experience. Section four describes the methods used in undertaking the study. The findings of this research are detailed in section five, followed by a discussion of implications and recommendations in section six. Section seven addresses the study's limitations and section eight concludes with an overarching summary of the study.

## 2. Research context

This research took place in Victoria, Australia, where care of clinically and psychosocially 'complex' and older patients forms a large proportion of hospital activity [6,10,11]. However, this type of care is not unique to the study environments. In the context of rising chronic conditions prevalence and ageing populations [12], nationally and internationally, hospitals are under pressure to deliver an effective, efficient service for the communities they serve [8,10,12,13]. Meeting the time-intensive care needs of so-called 'complex patients' is thus challenging [14]. As a result, the problem of healthcare complexity is often framed as a patient level issue, however, system complexity is equally problematic [15].

### 2.1. The complexity of complex care

In this study, a 'complex patient' is defined as one experiencing an entanglement of issues adversely affecting wellbeing [14]. Besides the physiological impacts of disease, the

wide-ranging adverse effects of complex ill-health can extend to a range of issues including interpersonal relationships, employment opportunities, health care affordability, community connectedness, mental wellbeing, sense of identity, and more [16,17]. In contrast, modern hospitals are organised to provide disease-centric, episodic health care and rapid patient throughput [18], making them less well adapted to the multiple, interconnected needs of 'complex patients' [10].

The complexity of the modern hospital compounds the issue of patient complexity, with many organised as distributed delivery systems [19]. This distributed design results in care delivery across different programs and locations according to patients' acuity and health conditions [4]. While intended to improve efficiency by providing patients with appropriately matched care, it also contributes to healthcare fragmentation. Patients often encounter multiple care teams within a single hospital stay, which can be disorienting and harmful for older 'complex' patients [4,10]. Healthcare fragmentation also gives rise to other issues, such as incomplete clinical handover [20], incommensurate approaches to care across different settings [21], and barriers to the progression of care [4]. The structure of the system thus contributes to patient harm. In Australia approximately one in four patients with an overnight hospital stay will experience some form of hospital acquired harm [22], in part through care fragmentation.

**2.1.1. Purpose of interprofessional care teams.** In response to the rising complexity of patient need, interprofessional hospital care teams inclusive of allied health professionals are now commonplace in hospital settings [4]. Interprofessional practice addresses the need to integrate care for the 'whole' person, while also responding to the ever-expanding knowledge base of contemporary healthcare [23]. Therefore, while the intersectional nature of health complexity calls for disciplinary diversity and expertise, at the same time, integration of disciplinary knowledge, skills, and practice is needed also [3,23]. An overarching goal of interprofessional practice is for collaboration to enhance efficacious, compassionate, person centred and inclusive care giving [4]. For this reason the calibre of relationships both within and between care teams has a large bearing on the effectiveness of interprofessional practice [4].

**2.1.2. Multidisciplinarity versus interdisciplinarity.** The term 'interprofessional' as applied in this study denotes both multi and interdisciplinary workforce models, while recognising also some key differences between these approaches. According to Jessup [24], members of a multidisciplinary team work together while each discipline adheres to its disciplinary practice scope. Patients are engaged separately by each discipline, team members contribute their knowledge to generate a global understanding of the issues at hand, and strategies are developed from a disciplinary perspective [24]. In contrast, an interdisciplinary approach integrates knowledge at the outset of patient care [24]. Different disciplines of a team work together alongside patients to identify issues and plan care [24]. Interdisciplinarity may also promote the integration of knowledge from different disciplines to generate emergent or hybrid professional roles [24].

**2.1.3. The experience of interprofessional practice.** Although interprofessional care is now common in hospital programs including in acute [13,25], sub-acute, and community settings [26], it cannot be presumed that knowledge sharing and integration happens simply by bringing together different disciplines to form a team [27]. Furthermore, it should not be assumed that a successful model of teamwork in one healthcare context it will work well in another [4]. Rather, care locations and patient cohorts have different characteristics that need to inform the makeup, activities, and practice norms of an interprofessional team [4]. In addition, across discipline socialisation and should happen in the process of implementing interprofessional care teams [4,27], since this type of care runs counter to the traditional

approach of disciplinarity [28]. Perceptions of disciplinary incommensurability [21] and professional resistance [27] will thus need to be resolved for a team to be effective.

Nevertheless, studies show that hospital based interprofessional practice can be an enriching experience for healthcare professionals. For instance, in their study of transdisciplinary health care, Cartmill and Soklaridis [29] identified a high degree of satisfaction among team members, a sense of shared, distributed power, and a supportive culture that engendered capability building and resilience in the face of challenging work. Likewise, in a study exploring the experience of early career hospital physiotherapists, Martin and Phan observed that participants enjoyed the pace of work and the learning that their environments engendered [30]. Still, the same study also pointed out several difficulties. These included a perpetual focus on patient discharge planning, the issue of conflicting institutional goals, and the challenge of providing a professional opinion that is at odds with organisational objectives [30]. It appears, therefore, that interprofessional practice is generally experienced as a complex phenomenon.

## 3. Complex adaptive phenomenology

CAP, the conceptual framework of this study, combines principles of phenomenology with 'complex systems thinking' [9,31]. Experience, which is conceptualised as a person's perception of, and psychic and physical presence in, the world is the central theme of phenomenology and constitutes one's "lifeworld" [30]. As Read explains, lifeworld is the manifestation of experience; it is the only world one can know and is thus uniquely individual [32]. The practice of phenomenology entails focused, 'intentional' engagement with the experience of a phenomenon both to reveal its nature, or 'essence', and to unveil what lies beneath what seems apparent and obvious [33–35]. CAP combines principles of phenomenology with a complexity-oriented framing of experience as a complex adaptive system (CAS) to create a holistic account of lifeworld [9].

CASs are considered to be ubiquitous; natural systems or social systems for example, are often described CASs [36]. They consist of many dynamically interactive parts [37] and as a result, they are said to behave 'nonlinearly 'and unpredictably [38]. CAS's are also evolutionary in nature; the pre-existing conditions of a CAS form the basis for how it might respond to perturbations within or external to its environment [39]. Because CAP defines experience in CAS terms, it is assumed that individual clinicians may respond differently to one another when exposed to similar phenomena. In other words, the evolving behaviour and open, interactive structure of experience make it unique to the individual [37]. Altogether, these features align well with phenomenology's characterisation of experience as a contextual and dynamically individual phenomenon [9,40].

By integrating CAS thinking with phenomenology, experiencing is conceived as a recursive process of perceiving and responding to sensory inputs of different forms and from different sources [41]. Experience is thus also considered to be 'lifeworld-shaping', because experience and environment are dynamically interpenetrating and interdependent systems [9,42]. This framing was applied to the analysis of study participants' interview data, to illuminate the interconnected structure of their experiences of hospital-based complex care [9].

## 4. Methods

The researchers consulted the consolidated criteria for reporting qualitative research (COREQ) as guidance for this study [43]. The COREQ framework consists of several intersecting domains: Research team and reflexivity, study design, and analysis and findings [43], which the following sections address.

### 4.1. Research team

As part of a larger research project, this study was undertaken across three comparable, tertiary health services in metropolitan Melbourne. The research team consisted of a PhD candidate (FB) whose professional background was complex healthcare, and a PhD supervisory team. Two supervisors (SL and HN) had a background in healthcare and research, and the other (VF), was an educator and researcher with expertise in organisational leadership. All team members contributed to the development of this article. In addition, principal investigators appointed at each study site assisted with participant recruitment.

### 4.2. Recruitment

Participant recruitment occurred commenced 15-11-2019 and was completed 28-4-2021. Recruitment entailed a purposive approach [43], with site-based principal investigators matching participant selection to the study's aims [44]. The participants' details are summarised in Table 1.

Two senior physiotherapists and one senior social worker were recruited. Pseudonyms have been used in place of actual names to preserve participants' anonymity. One physiotherapist is identified as Jane, the other as Helen, and the social worker as Fiona. Jane worked as a multidisciplinary team leader in an acute general medicine program, Helen led an interdisciplinary team in an ambulatory care setting and Fiona was the lead social worker in a multidisciplinary subacute in-patient setting. Altogether, the participants' differing roles and settings of care provided an appropriate level of complementarity, as well as variety, to inform the study.

### 4.3. Ethics

Ethical approval was granted by the lead health service (The Alfred Ethics Committee, Project 52394, Local reference: 375/19), and study site ethics and governance committees (SSA/52394/MonH-2019-185017 and SSA/52394/ph-2019). In view of the imposts of the coronavirus pandemic, two revisions were made to the study protocol; one to include telephone and video interview methods, and another to extend the timeframe allowed for data collection. Participants were each given plain English written and verbal information about the study, inclusive of how participant privacy would be protected and how the study data would be used in this research. Each participant provided written informed consent to participate in accordance with l4.4 legislative requirements [45].

### 4.4. Data collection

FB conducted in-person interviews with each participant in a private setting. The interviews were of 40 to 50 minutes duration. Semi-structured interview questions explored participants' perceptions of interprofessional complex healthcare, contextual factors affecting caregiving practices, and interpersonal interactions with colleagues and patients (see supplementary file). The recordings were professionally transcribed, using a naturalistic approach to capture

**Table 1. Participant details.**

| Pseudonyms | Discipline | Role | Practice location | Years experience |
|---|---|---|---|---|
| **Jane** | Physiotherapy | Team leader | Acute: General medicine | 7 years |
| **Helen** | Physiotherapy | Team leader | Subacute: Ambulatory | 6 years |
| **Fiona** | Social work | Senior social worker | Subacute: In-patient rehabilitation and aged care | 28 years |

nuanced expressions and emotions in the dialogue [46]. Each participant was sent their transcript for member checking to ensure the accuracy of recordings and to enhance the study's validity [47], however no feedback was received. The researchers therefore had to assume that participants were satisfied their transcripts were accurate.

### 4.5. Analysis

An initial reading of each transcript was followed by a thorough line-by-line analysis, in accordance with the methodology described by Walker [48]. Following several weeks of reflective journaling and discussions with the supervisory team, a second analysis was completed in which the study's findings were distilled. Throughout, CAP focused analysis on the entangled interplay of contextual and interpersonal elements shaping each participant's lifeworld [9]. This approach sought informational enrichment, to illuminate the expansive horizon of participants' experiences [9]. Finally, awareness of the unique nature of experience informed data analysis [49] while overall, the study also revealed findings that are likely to be 'typical' within complex care practice settings [50]. Table 2 summarises findings from both analyses. The detailed elements of the first analysis informed the integrated 'essential' categories in the second analysis. A detailed account of CAP's analytic method has been given elsewhere [9].

### 4.6. Reflexivity

Researcher reflexivity was employed throughout this research to strengthen the reliability and trustworthiness of findings, especially as two of the study's authors have a background in health care. Reflexivity entailed self-critiquing assumptions and prior knowledge of the subject matter [32,40] as well as two separate instances data analysis, coupled with reflective journaling and feedback from the supervisory team [40]. However, the researchers' professional background in healthcare has also strengthened this study. Besides creating heightened awareness of the risk for bias [51], prior knowledge within the research group allowed for a nuanced understanding of the subject matter and the adoption of a co-constructive attitude in the development of findings [49]. A diversity of perspectives within the research group also fostered an expansive account of interdisciplinary complex care.

## 5. Findings

The study's findings will now be discussed under the following headings:

• Workplace culture, leadership, and empowerment

• Interprofessional practice

• Healthcare ethics

• The ambiguity of complex care

**Table 2. Findings of analyses.**

| ANALYSIS 1 | ANALYSIS 2 |
|---|---|
| Care context, culture, power, and clinical leadership. | Workplace culture, leadership, and empowerment. |
| Interdisciplinarity and integration of care. | Interprofessional practice. |
| Meaning and purpose, validation, and person centredness. | Healthcare ethics and values. |
| Complex patient attributes, fragmentation of care, work and productivity. | The ambiguity of complex care. |

## 5.1. Workplace culture, leadership, and empowerment

This research found that each participant's workplace had a unique character that was informed by the organisation's location, size, work pace, and focus of care. Each environment dealt with different kinds of patient issues along the continuum of acuity: acute inpatient, subacute inpatient, and subacute ambulatory. Yet despite these differences in contexts, the participants' approach to practice had much in common, especially regarding teamwork and leadership, as the following passages reveal.

Helen's organisation abutted a semi-rural setting, and although it was the smallest of the three study sites it still spanned multiple geographical locations. Helen worked at the organisation's tertiary site, leading an interprofessional team in a sub-acute ambulatory care program. For two days per week she undertook administrative work and for the remaining three weekdays she was engaged in clinical practice.

Helen described her colleagues as "community focused", less concerned with "climbing the ladder to the CEO", and "…just happy to really do the best you can for the day, and then go home and spend it with your family." She advised that her team routinely shared clinical decision-making, and felt this contributed to a high standard of patient care [52]. In addition to the team's 'distributive' decision-making approach [53], Helen employed a particular style of team leadership that Cooksey [54] terms 'learnership'. She modelled and fostered collaborative and adaptive learning with her team in response to the complex care needs of patients. Mutual learning and leading thus occurred in tandem [55] which Briggs and McElhaney [4] explain generates the "knowledge generating dialogue" that is essential for effective team-based complex care.

However, Helen explained that her approach to leadership had been developed through trial and error rather than formal training. She advised, "… I came into the role quite inexperienced… often clinicians get given these leadership roles and they're not businesspeople and they've not had leadership roles before". Furthermore, although formal supports were ostensibly on hand to assist Helen in developing her leadership skills, this research found organisational constraints made accessing them difficult. Helen explained,

> "We're all busy…Do I need to bother that even more senior person? I don't want to show that I'm not able to do it…the support, it looks like a lot on paper, but in reality, half of it goes by the wayside."

As a result, Helen developed her understanding of leadership by reading hospital policy and procedural documents, coupled with day-to-day practice and continuous internal dialogue and reflection. Helen's experience of early leadership also influenced her decision to differ in the approach that she took in developing others. Having shied away from seeking support herself, she held the goal of empowering team members front of mind:

> "I want to empower people to have the skill themselves… those skills in how to clinically lead, how to bring out the best, how to support and teach your team, they're the things that I've lacked training in and had to learn by doing it the wrong way initially".

From the perspective of CAP, Helen's response to her experience was interesting, in that it was resistive to the constraints that she had felt in her leadership journey. In the language of complexity science, her experience had prompted a 'bifurcation' in the path that she chose for her team [39]. As Byrne explains, bifurcation denotes the transformation of a system from one state to another [36]. Such transformations inform future behaviour [39], which for Helen entailed a more engaging and proactive approach to leadership with her team than she had

experienced. She wanted to be an 'available' leader who was also open to learning; she reached out to others rather than waiting for them to reach up to her. It was an 'emotionally intelligent' approach to leadership [56] that mitigated the sense of vulnerability invoked by having to ask for help from someone more senior.

Turning to Jane, her organisation was the largest of the three study sites, entailing a network of locations including three tertiary hospitals and multiple smaller sites. Jane led a team of allied health professionals practicing across several inpatient general medicine wards. Her role entailed clinical supervision, governance leadership within her program, and consultation for other medical programs and a private wing of the organisation. Like Helen, Jane also held a clinical caseload which occupied approximately half of her time, depending on service demand. Again, similar to Helen, Jane placed a high value on shared decision making, describing it as "a massive thing to bring that to the team." She explained, "I love it, because you want to feel part of the team, you want to know you have your role to play, that you can help…shape this person's care…everyone wants to feel appreciated and acknowledged."

For Jane it was important to her profession and patient care that early-career clinicians were confident to contribute to interprofessional discussions, but in her experience the thought of joining a clinical conversation could seem "daunting". Jane explained that she "worked hard" to empower junior colleagues to contribute to clinical discourse, describing this challenge as follows:

"I think the funny thing is that whilst it's great when you experience that and you're confident with it, on the other end of the spectrum it's a really daunting position to be in, and probably just easier if someone didn't really take much notice of you… to turn around and talk to a reg (medical registrar) or a (medical) consultant and say what you think or question them or adding your two cents isn't easy, and so I think for junior staff it takes a lot of coaching and development for them to have the confidence to speak out in a team".

The hesitation that Jane witnessed in junior clinicians' willingness to speak up in a team context, especially if it involved presenting a contrasting view or contradicting a doctor's opinion, highlights the constraining effect that perceptions of hierarchy can have on team functioning [57]. However, this finding and its implications were not unique to Jane's setting. The third participant, Fiona, encountered similar issues.

Fiona was the senior social worker in an inpatient subacute setting of a large, inner city, multisite health service. She worked across several programs including rehabilitation and aged care, all located at the same hospital site. In addition to her clinical work, Fiona was responsible for the professional supervision of junior social workers and, like Helen and Jane, she was committed to developing and empowering them. However, Fiona also found junior social workers to avoid engaging in multidisciplinary team discussions. To address this issue, Fiona held staff-led education sessions, conducted reflective practice sessions, and used humour to help junior social workers feel relaxed and safe in the team environment.

In addition, being familiar with the kind of information that senior management expected from staff, Fiona coached team members to have well-prepared responses to predictable questions, using a clear communication structure. This finding was interesting in terms of how it might have influenced clinical practice, as the approach appeared to be implicitly procedural and routinised. Because of the pervasive focus on discharge planning in hospital settings [30] this rehearsed response raised the question of whether it served organisational goals or those of patients and families. Comprehensive discharge planning is important, but when throughput and productivity become its end goals the needs of complex patients risk being reduced to what is achievable in the short term. This then limits the sustainability of care that is provided [8,30,58].

The more obvious finding, however, concerned the discomfort that junior clinicians experienced in having multidisciplinary discussions. One possible consequence of this is that the perspectival variety that interdisciplinarity ought to bring to a care setting is muted by team member assumptions about their place in the team's social order [57]. Such adherence to an implicit team hierarchy risks limiting a team's overall capability [59]. The participants' attempts to engage junior clinicians in participation demonstrated the importance that they placed on professional diversity in complex care [3].

With respect to the participants' perspectives on team power dynamics, these appeared relative in nature. For example, Fiona had felt unsupported in her pervious role which she contrasted with her present workplace that had an inclusive, "can-do" culture: "I'm supported, and the culture, I'm appreciated, not just by my own discipline, but by others…I feel like I'm able to offer and that they can accept. It's like an equal relationship…I think that's just amazing."

However, Fiona's experience of workplace satisfaction was not derived solely from collegiate appreciation. Fiona also found her professional values to be reflected in her workplace, and this appeared to invoke in her a sense of connection and loyalty [60]. Like her colleagues, Fiona was motivated by a commitment to overcoming obstacles in patient care, and she advised, "No matter what the problem is, even though it looks like the most unbelievable, unable thing that we can do…I'm surrounded by this culture". A workplace reflecting shared ideals gave Fiona confidence in her approach to leadership, as it mirrored that modelled by other leaders in her sphere [61]. Fiona thus experienced leadership as a 'collective' phenomenon [61], which bore similarities to Helen's experience.

Jane was equally appreciative of her team and the collegiality within the general medicine program. However, like Fiona, she compared this to interactions in a previous context to suggest that inclusivity was not universally practiced in hospital settings:

> "I'd worked in (a surgical specialty) for a bit and I remember the surgeon, you'd say, "But they (the patient) can't go home. They can't walk", and the surgeon would say, "Well, I've fixed their hip fracture, so I'm done", like, "I don't care what happens from here".

Jane interpreted the dismissive nature of this interaction as devaluing [60]. In fact, because interprofessional practice depends on teamwork, these kinds of trivialising interactions run the risk of lowering the quality of patient care [62]. Jane contrasted that encounter with the favourable culture in general medicine and attributed the difference in attitude to a shared appreciation by generalists that interprofessional practice was essential to effective complex patient care [4].

Similarly, Helen compared the collegiality within her team with the hierarchical manoeuvrings that friends working in other health services had experienced. Maybe, working in the ambulatory setting, at arm's length from the high pressured inpatient environment [63], contributed to this difference in workforce behaviour. It could also be that organisational size may have had a bearing, with decision-making power in large organisations still mostly vertical and centralised [64]. A centralised power structure is known to induce work units to strive for autonomy over local decision-making, which can manifest as workplace tension [64]. In contrast, collaboration and shared decision making involving the expertise of all team members has been shown to offer a more rewarding experience of power [64,65]. Likewise, among the participants, it appeared that power sharing was perceived as strengthening both individual and team performance, with interactions between individual clinicians contributing to the 'whole' of care [66]. Overall, however, the complexity of patient care seemed to be the key driving force for a collegiate culture in each study setting.

## 5.2. Interprofessional practice

As just mentioned, the participants' narratives highlighted the need for interprofessional practice in view of the complexity in patients and the healthcare system [4]. According to Fiona, the situations she encountered could easily overwhelm less experienced clinicians. Recognising the challenging nature of complex care as a risk for clinician stress, burnout [67], and health care quality [63], Fiona tried to create a psychologically safe environment for her multidisciplinary team by ensuring problems were discussed in a supportive manner [63]. Yet, despite Fiona's seniority, she found it challenging to engage clinicians in such discussions.

> "I think that you have to breed, promote, and encourage safety…how do you get a team that comes together to actually talk openly about what they are uncomfortable about, what they're struggling with as clinicians when you have a variety of skills in the room?… There are gaps in our knowledge and to talk about gaps…in a group environment is actually quite overwhelming."

Interdisciplinary discourse is known to be an intimidating prospect for some clinicians [24], yet knowledge-sharing is essential to forging strong team interdependencies and closing knowledge gaps [59]. In fact, interprofessional discourse allows a team to evolve, respond flexibly to emergent issues, and cultivate a tacit-knowledge base, all of which are needed for complex healthcare in a complex system [59]. Therefore, in Fiona's mind, the value of group discussions outweighed the perceived risks of this by individual team members. However, in order to mitigate psychological distress [63], she had created 'rules' for engagement in these discussions and regularly invited subject matter experts to share their knowledge with the team.

The need for such investment in team development was evident in the complexity permeating the study settings. Complex conditions exert a cumulative effect, to adversely impact numerous aspects of patients' lives [17] while, like much of Western health care, the systems participants had to navigate were also complex [68,69]. The field of complex health care was thus infused with 'wicked problems' that were individual, multifactorial, and lacking in clarity [8,70,71]. However, as 'wicked' problems are without clear solutions, health complexity appeared antithetical to the standardising logic of 'evidence based medicine' that underpinned clinical practice [72]. Amid this paradoxical situation, this research found that the participants had become masterful at adjusting care to the vagaries of health complexity. They had learnt to combine a heuristic approach [73] with formal knowledge and "inter-individual interaction" among the care team [66].

While heuristic reasoning commonly implies biased, "fast, automatic, and non-conscious" reasoning [74], the participants' heuristic practice instead mirrored Beer's [73] description of it as akin to climbing a ladder that extends into a dense fog. Each rung is assessed for reliability before leveraging it to access the next [73]. Such decision making resembles a careful titration of strategies rather than a sprint toward a solution [73]. The participants' heuristic practice was finessed, as per Hopkins and O'Neal [53], by continuously asking questions and positing options. The following narrative exemplifies Helen's 'heuristic' response to a shifting account of patient need:

> "Yes, (the patient has) a sore knee, but they're also isolated, and so fearful of moving, or fearful of things going wrong with them, medical things, they've a past history of things not performing according to the textbooks. So, they've got all this worry about 'oh, it's bound to be another failed bit of therapy' so their belief in their recovery is low and they've got lots of things that they are dealing with; they live alone, they don't feel supported… So, you're dealing with a completely different set of circumstances (than the single issue of a sore knee)".

With past, present, and assumptions about the future informing the patient's state of self-belief and ability to engage in health care [75], Helen's approach was first to listen to their unfolding life-story in order to grasp what mattered to them. Her objective was to "collaborate on a good plan that works for you (the patient)". She tailored care so that it made sense for the patient and herself. Moreover, the objective of care did not need to be singular. For instance, if a patient needing physical therapy also experienced loneliness, Helen would suggest group therapy sessions, to tackle both issues concurrently. She thus flexibly adjusted interventions to be both recuperative and preventative, using a synthesis of formal, collaborative, and experiential knowledge.

Like Helen, Jane described her practice as "treat what you see" rather than "getting caught up in what you're expecting to happen". Experience had taught Jane "there's a very blurry grey kind of area that exists in general medicine and with patients who have so many bits and pieces wrong with them". Jane was referring to the unpredictability inherent in complex health care that renders health complexity resistive to linear problem solving [76].

Instead, Jane explained,

> "I think my problem is this and I treat it with that and it's improving, that's pretty good evidence… that's anecdotal evidence that this is improving, but at the moment that it stops (improving), you need to find something else".

In other words, Jane grounded her theoretical 'evidence based' knowledge in a 'real world' context by working with patients according to their individual circumstances [77]. In fact, this was each participant's approach to care. Their empirically grounded practices involved action-based 'tacit' knowledge that they had developed as a result of experience [78]. Tacit knowledge develops through immersion in a social process [79] and the participants each spoke of their access to the shared knowledge that was emergent of their interprofessional 'niche' [42]. Collaboration, familiarity, and team discussions allowed sense to be made of 'wicked problems' that were peculiar to each patient in each setting [4]. The participants valued and had normalised this form of emergent knowledge creation, as demonstrated in Jane's statement below:

> "…we do need the healthcare team, so with the multi-morbidity patients and the complexity that exists in general medicine, you do need each member of the team to bring that next piece of the puzzle to kind of put it all together."

### 5.3. Healthcare ethics and values

Just as patients are the embodiment of their experiences [75,76], so too were the participants, with experiences shaping their values, motivations, and perspectives in clinical practice. Some events can leave a lasting impression, however, as Jane disclosed in the following account of a patient whose wishes went unmet by the care team.

> "I think I was quite junior at the time, and I just thought it was so jarring that someone could be so affected by the recommendations made to them that they were wanting to take their own life… She didn't feel like we were listening to her, and we didn't feel like she was listening to us, and she was so upset and in the end her health took a dive…she ended up passing away in that hospital stay, and I think sometimes you just think, what is patient centred and what is person centred?"

This scenario highlights the relational nature of healthcare, that while between a practitioner and patient, reflects the clinician's relationship with wider society also. As Li et al.

[80] advise, "the soul of caring is in formulating human relationships to 'care about and care for' the patient, the family and society at large". However, institutional policies can override person centred care [81] to influence ethical decision making and impact patient sovereignty [82]. Experiences like that described by Jane are also consequential for clinicians, as they are felt, remembered, and as such, embodied [37]. Such events can lead to moral distress and potentially, the attrition of healthcare workforce [83]. Yet, while acknowledging the effects of distressing encounters on patients, clinicians, and others [17,30], the participants explained they were an inevitable part of complex care practice.

Hence, through her experience of working in ethically complex, risk infused environments, Fiona had learnt to accept patients' choices, even when these appeared unwise. She reasoned that patients lived their lives in the community, with the hospital only "a transition" point. That said, complex patients often 'transition' to suboptimal circumstances following their hospital stay, especially those that are vulnerable, frail, or elderly [4]. Fiona was pragmatic, however, reflecting that large scale issues like social health inequity [84] were beyond her ability to influence. She explained, "you can't separate health and be asking health to be responsible for all the social determinants...I think that that's quite unfair and I think it's a disproportionate expectation for health".

According to Fiona, a more contentious issue concerned seeking a patient's permission for disclosure. She expressed a deep respect for patients' private stories and did not assume these were hers to access simply because she was a health professional. Fiona saw permission seeking as a critical step to rapport building which, in turn, was essential in the delivery of effective complex care [4]. In this regard, Fiona felt that healthcare professionals had much to learn:

"We assume the person will disclose because they are in a hospital and it's on our terms. So, we've already started on the back foot. (The patient) might be really, really sick and they've lost their sense of independence, but I have to ask for permission and that's it, and I don't think we do that".

However, rapport building, learning about patients through their stories, and finding out what matters to a patient requires an investment of time [72,85] which was a scarce resource in Fiona and Jane's fast-paced work environments. In contrast, Helen's ambulatory setting appeared more conducive to building rapport and addressing the social determinants of health. Interestingly, however, Helen seemed unconvinced about the value of her work. She explained her reservations by way of an experience that she considered analogous to complex care. It concerned a time where she had gone against the advice of colleagues and had organised for a piece of old gym equipment to be repaired. Subsequently, the equipment was deemed unsafe and had to be discarded. Helen explained,

"I use that analogy. Is that (treatment)…the best way to help that patient?...in the context of that person, where does it fit in and what's the bigger picture? …From having that experience, 'don't fix the gym equipment when the whole gym equipment was not worth fixing', (it) made a big impression".

Helen thus likened the 'complex patient'[14] to faulty gym equipment, a characterisation that seemed out of keeping with previous statements as it signalled a stereotyping perspective. Stereotypical views of complex patients are known to permeate healthcare and negatively impact both the clinician and patient experience of complex care [14,86]. There is also evidence to suggest that implicit stereotyping of patients often goes unrecognised, despite it being influential in treatment decision-making by practitioners [87]. Implicit bias

is particularly concerning in the case of complex patients that are already disadvantaged through chronic disease and psychosocial disparity [87]. Hence the term 'corrosive disadvantage', coined by De-Shalit and Wolff to denote the effect of inherent bias when it further disadvantages those already disempowered [87]. Helen's reflections highlight the subtle nature of implicit bias in its capacity to shape a clinician's perceptions and, potentially, practice [87]. Recognition of bias in one's mindset and practice is thus an important safeguard in patient care [86,88].

## 5.4. The ambiguity of complex care

As highlighted in the literature [89,90], explaining the value and nature of generalism to the medical community has its challenges. Some of this difficulty lies in the evidence base for generalist medicine that is dependent on specialist medical knowledge [90]. However, generalist practice is equally reliant on 'real world' evidence; in fact, this may be its defining feature [77]. The term 'real world' draws attention to the complexity of human experience that does not lend itself to a standardised account [36]. As a result, the generalist approach to practice is to inductively tailor evidence-based specialist guidelines to the individual needs of patients [77]. Such tailoring is necessary because most clinical management guidelines are disease specific, and so not directly applicable to the multimorbid patient [91,92]. As a result, generalist practice involves a degree of uncertainty that can be off-putting to some clinicians [92]. Lynch et al. [77] concur that a human preference for simplicity and certainty over complexity informs the medical paradigm [93]. This phenomenon is highlighted in their description of medicine's "…remarkable reluctance to acknowledge the integrating force of the generalist gaze" [77]. The capacity and willingness to work with uncertainty thus requires a complexity oriented aptitude in addition to experience and expertise [77]. This also means that the support and guidance of generalist experts is critical for junior clinicians [93].

It was unsurprising therefore, in view of the complexity of generalist practice [93], to find the participants considered generalism to have an image problem. Jane observed that colleagues working in specialty healthcare failed to see how she could enjoy caring for people with chronic and complex conditions. She described generalism's image as problematic,

> "… not just for patients but for staff as well…it's often seen as an easy field…but how do you go about explaining that to someone? Gen Med certainly doesn't have, like, a sexy image. I would have colleagues who probably think I just drag a few grannies around all day."

Jane's reflection also suggests that ageist stereotypes inform perceptions of generalism, a suggestion that the literature supports [88,94]. As to generalism being an 'easy' field of practice, Jane rejected that suggestion, stating,

> "…these days there's so many people coming in with so many morbidities, that it just makes things so much less straightforward, and having people understand how they all interact and how the care providers interact is just a massive challenge… going forward I think it can only get harder as we have an ageing population and as public funding is more and more stressed, I can only imagine it's going to get more difficult".

Time and resource constraint in the busy in-hospital study settings added to the difficulty of providing effective complex care [95]. Constant pressure for hospital beds caused Jane and her team to direct patient therapy toward hospital discharge to the extent that discharge had become a superordinate care-team goal. This research found that despite Jane's influence in

care planning and decision-making, the pressure of hospital demand was ever-present. She advised:

> "…Some of the challenges around acute hospital care at the moment revolve around time and money, essentially resources, and I think, unfortunately, that does impact on patient care…as soon…as someone is well enough, we're trying to discharge them home…everyone's on a time schedule, everyone's…got more work to do than they've got time…that then unfortunately, causes this kind of lack of communication amongst the team, but also with the patient, and it's not good for patient experience.'".

Fiona also felt the effects of 'system pressure' although in her work this arose from systems and services beyond the hospital walls. Much of Fiona's role entailed transitioning patients from hospital to community-based care. She explained that accessing community services had become increasingly difficult because standardised and inflexible service eligibility criteria were progressively replacing clinician-driven decision-making [for example see, 96,97]. This made negotiating a supportive discharge for patients struggling with their loss of independence even more difficult. Hence, although the problems facing Fiona were 'social' rather than clinical in nature they still added to the ambiguity of complex care and required expertise to resolve [93].

In contrast, Helen and her team were well placed to "knit together" patient care, "not perfectly", but to the extent that both social health determinants and physical health needs could be addressed [26]. This was enabled by the fact that Helen's ambulatory service was part of a government program charged with promoting patients' health independence [26]. Team interventions were thus focused on delivering person centred and sustainable health outcomes to a greater extent than the fast-paced in-hospital setting allowed.

Variation found in the participants' experiences of health complexity highlights how institutional drivers shape the way care is provided. The perceptions and actions of allied health clinicians, like the participants, inevitably inform how the healthcare system deals with 'complex patients'. This study found that the inherent meaning of institutional key performance indicators, systems, and processes was instilled in the participants' motivations for practice [98]. Therefore, the environment of clinical practice was *formative* of the clinician patient relationship. With environment integral to the clinician patient relationship, the participants had little option but to submit to its constraining effects in the in-hospital settings and benefit from its enabling effects in the ambulatory space. This finding illuminates the interconnected and insidious nature of institutional power in the clinical setting. It calls for healthcare decision-makers to be aware of its effects at the coalface of care, and to balance its advantages against its unintended negative consequences.

## 6. Discussion

A review of the literature found little addressing the experience of allied health professionals working with 'complex patients' in interprofessional hospital teams despite that the need and prevalence of this practice is increasing [5]. Some scholars have attributed this gap in the literature to an overemphasis on health care efficiency, suggesting that this overshadows the need for clinician meaning and purpose in clinical practice [45,99]. Some also propose that a lack of attention to the clinician experience of caregiving has contributed to workforce attrition and looming workforce shortfalls [45]. It is important, therefore, to learn from clinicians about how they experience complex care practice in hospital settings in order to address barriers to meaningful work and retain a skilled allied health workforce.

In Fig 1 the findings of this research are displayed according to four domains that held commonality for the participants. Intersecting each domain is the issue of 'patient and system complexity', two features that were inextricably entwined with the participants' lifeworlds. Complexity was fundamental to the participants' experiences of hospital-based complex care, while workplace leadership and culture, collaborative interprofessional practice, issues of healthcare ethics, and the ambiguity of complex care, exerted mediating effects. The impacts and implications of these findings are discussed next.

Despite working in different settings and models of care, this study found the participants shared a key value that the researchers term 'empowerment of self and others'. Each placed a high value on collaborative practice and was vested in developing less experienced clinicians to be active contributors to interprofessional dialogue. The participants' mutual focus of clinician empowerment highlighted the level of importance they placed on intellectual diversity in complex clinical practice [3]. Given the variety of problems in the complex care context, it was also clear that the junior workforce needed such support and development [93]. Yet, although support was on hand, it seemed junior clinicians were still hesitant to participate in interdisciplinary discussions.

Brown and Yu [100] reported similar issues in their study of occupational therapy students' experience of clinical placement. The authors approached the subject of clinician confidence through the lens of resilience, and they critiqued undergraduate education for a perceived lack of investment in developing resilience in students. It was argued that resilience was an essential capability for effective teamwork and for managing time pressure, multiple competing demands and emotional stressors. Reflective practice was offered as a resilience-building measure to counter the negative effects of a stressful work environment [100], an approach that Fiona had likewise adopted in her practice. Similarly, a study by Martin et al. [30] found physiotherapy graduates to be underprepared for the complexity of patient care and to lack the negotiation skills needed for interprofessional practice.

Thought diversity in an interprofessional context manifests in dialogue and actions, each helping to make sense of the ambiguity and uncertainty encountered in complex caregiving [4]. However, tensions can arise because of perspectival differences, clinical reasoning may be challenged, and cultural differences between disciplines may surface [100]. For this reason, the

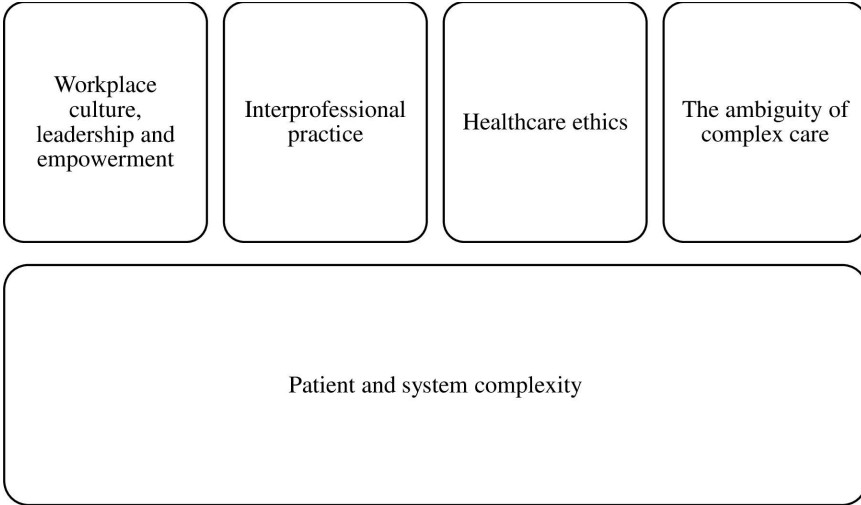

**Fig 1. Factors that shaped the participants' lifeworlds.**

participants coached junior clinicians in how to negotiate various tensions in a manner conducive to effective complex care team work. Clinical coaching was thus an essential leadership skill that was used to transform inexperienced and hesitant clinicians into active contributors to interprofessional practice [101]. However, it appeared that the participants had adopted this approach organically rather than through formal training..

Piers et al. [101], noticed a similar gap in allied health leadership development in their study of interprofessional teamwork. It therefore warrants considering whether this is the case broadly and if so, how institutional decision makers might support emerging allied health leaders through targeted preparation. Relatedly, Bradd et al. [2] have pointed out a gap in the literature regarding allied health leadership development. In addressing the issue, they created an in-house allied health leadership development program that included theoretical content, action learning sets and leadership coaching. The results of their study showed the program made statistically significant improvements in the participants' leadership capabilities and confidence. The findings demonstrated that organisational investment of this kind can reap benefits for clinicians, patients, and health services. In view of the findings of this research, leadership programs like those developed by Bradd et al. [2] appear necessary and show promise to be applicable more broadly. The fact that allied health leadership development appears to be an under-researched topic [2] adds weight to this suggestion.

In addition to coaching, the motivation of 'empowerment of self and others' was also found to influence the participants' leadership styles that may be best described as 'distributed' or 'shared' [59,102]. Distributed leadership complemented team-based sense-making, with both shaping how power was enacted, problems were explored, and decisions were made in relation to patient care [79]. Distributed leadership contrasts with hierarchical leadership that incrementally reduces autonomy at each descending rung of the hierarchical ladder [53]. Hierarchical leadership has a long history in healthcare, its adoption coinciding with the social transformation of the industrial revolution [53]. In line with mass production and the mechanistic social order of the time, hospitals had begun to operate on a large scale [53]. Today, the industrial scope and hierarchical order of hospitals exists in tension with the goal of person-centred healthcare which frames every clinical encounter as an opportunity to address patients' individual needs [76,77,103]. The participants' approach to leadership thus stood out because it embraced the notion of shared decision making to ensure individualised centred care, thereby differentiating itself from embedded tradition [53].

Nevertheless, this study found that institutional hierarchical order had a particular bearing on Fiona and Jane's study settings (Jane and Fiona's). While both participants enjoyed a collaborative workplace atmosphere it was clear that 'top down' measures drove the pace of work and perhaps, junior staff's hesitation to involve themselves in decision making. This form of control appeared ill-matched to the needs of complex patients [104] mainly because hierarchical leadership is contingent upon standardisation in roles and practices [64]. Standardisation places controls over the behaviour of a workforce under the assumption that institutions function in a predictable manner to deliver predictable outcomes [64]. However, while a hierarchical structure may function well in a stable environment [64] it is less well suited to complex care contexts where uncertainty is commonplace and where flexible, individualised decision making is essential [73]. The lack of a 'recipe' for complex healthcare explains why Heath [76] describes it as "messy". The nature of health complexity escapes a static conceptualisation of health and illness making it a necessarily bespoke domain of practice [76,105]. In response, the participants' approach to leadership modelled 'empowerment of self and others' so that junior clinicians would develop the skills and confidence to be effective interdisciplinary practitioners [4], because as Jane stated, "we do need every member of the team".

Yet, this study also found that, often, clinicians were not comfortable with uncertainty and at times, neither were the participants. For example, Fiona observed that seemingly unsafe patient choices challenged the inherent drive of her colleagues to "do the right thing" [106]. In a similar vein, Jane thought back on a time when her healthcare team and a patient had failed to agree on goals for care, which Jane then associated with the patient's subsequent demise. The event caused Jane to question the concept of 'person centred care'. Moral ambiguity of this nature can inflict psychological harm, demotivate clinicians [10], and contribute to burnout and workforce attrition [83]. In addition, Helen disclosed a sense of ambiguity toward her practice when she questioned the value of complex care. She compared complex patients to broken gym equipment and wondered whether these kinds of problems could or should even be 'fixed'.

Yet, ambiguity is an inherent feature of health care [93] and especially, complex care [76]. As a result, in today's context where patient complexity is common place, learning to accommodate uncertainty seems a skill essential to clinician performance and wellbeing [93]. It is suggested, therefore, that the socialisation of health professionals to uncertainty should begin at the undergraduate level of education for it to be a normalised and well-practiced aspect of health care, rather than having it set aside as an aberration [93].

Along these lines, this research found that the benefit of standardisation has limits and that rigid adherence to predictable pathways in clinical practice can be detrimental to healthcare quality. For example, Jane's experience in a surgical setting highlighted the linear thinking shaping a doctor's inflexible application of 'evidence-based practice' [4] and top-down decision-making [64]. It made no sense to Jane that a patient would be discharged from hospital if unable to walk, even if their hip were surgically 'fixed'. In contrast, the doctor in this scenario viewed the patient's trajectory as having deviated from the expected postoperative pathway, an outcome they found unacceptable. However, variation is often viewed as the enemy of 'stable' simplicity, a normatively desired state [104], especially in healthcare [15]. In this regard, Briggs and McElhany warn that clinicians ignore complexity at their, and patients', peril [4]. Adherence to evidence based practice is inarguably important for the safety and quality of patient care [107]. However, it is equally important that the application of such practice is grounded in a patients' 'real world' circumstances. Not doing so is likely to lead to failures in care [77].

Another finding of this study was that exposure, practice, and further education in Fiona's case, were key ingredients for building deep knowledge and capability in complex care practice. The participants had come to view complexity as an inherent and, in fact, interesting feature of their work. The integration of variety and unpredictability into every day practice was evident in their well-developed tacit knowledge base [108] that rendered formal knowledge more refined. Their application of tacit knowledge had an anticipatory quality that informed a holistic assessment of patient need in Helen's practice, nuanced the conversations that Fiona held with staff and patients, and underpinned Jane's attitude of "treat what you see" without the conviction such treatment would always succeed. Working with complexity had taught the participants to expect and adapt care to the unexpected, and to 'respect' and normalise this.

Nevertheless, compared with Helen's setting, Jane and Fiona's workplaces appeared to be less tolerant of flexibility. This was problematic in view of the issues they confronted which could be described as "wicked" [8,83]. Wicked problems are infused with interdependencies and unpredictability [15] which makes them behave as a "moving target" [109]. For this reason Sturmberg and Miles [15] advise that acting and then sensing for a response to an intervention rather than relying on 'fixed' rule-based solutions is the more effective way of dealing with wicked problems. Correspondingly, Fiona described the complexity infusing complex care as potentially "overwhelming", particularly in view of the organisational pressure

to progress patient care and the numerous barriers in place to achieving this objective. Jane also advised that clinicians always had more to do than there was time for. There thus appears to be value in examining the degree to which inflexible, top-down control in contemporary healthcare systems (for example, processes, eligibility criteria, and key performance indicators) adds to complexity by limiting clinicians' capacity to work adaptively [73]. In this regard, Helen's work setting provided the contrasting example among the participants. Her service, while still goal oriented, had adopted a flexible approach so that the team could address patients' emergent needs also.

Variation in the participants' experiences suggests that the different policy settings across the study sites were informed by different values, with some more reductive and transactional than others [77]. As one of several Government programs addressing the underlying issues of 'hospital demand', Helen's program aimed to maximise patients' long-term health outcomes and reduce their reliance on public hospital services [26]. In contrast, Fiona and Jane's inpatient environments bore much of the brunt of the 'hospital demand' issue. Under such strained circumstances, it is understandable that efficiency has become a key concern for governments and hospital administrators [8]. However, while efficiency measures can take various forms, in health care they tend to be financial in nature [8]. The reduction of 'efficiency' to economics has been identified as influential in how healthcare is provided and how clinicians experience caregiving [8,110]. In view of this connection, Lissack [104] warns that reduction cannot subsume complexity; rather, reduction augments complexity by failing to account for inherent variety. In a healthcare context, this means the perceived gains of applying simple solutions to complex problems can exacerbate issues rather than resolve them [8], and this seemed evident in the in-hospital study settings.

Because Fiona and Jane's practice contexts were highly exposed to the strain of hospital demand, "patient churn" [10], and institutional constraints [111], it appeared that they experienced the amplifying effects of reduction more keenly than Helen. However, the high productivity that was expected in Fiona and Jane's settings does not automatically deliver high quality or sustainable health care [8,110]. When the notion of efficiency is reduced to economic measures, the values shaping meaningful clinical practice may be eclipsed [7,8,110]. This can, in fact, lower the quality of care and thus its sustainability also [63]. In short, excessive economic constraint exerts 'top down' hierarchical regulation over clinician autonomy, decision-making, and satisfaction, with disempowering and potentially disenfranchising effects [7,110].

This is not to suggest that the care provided in Jane and Fiona's contexts lacked skill, compassion or, in fact, joy. Rather, the findings of this research suggest the participants' experiences were influenced by variation in the definition and primacy of efficiency at each care setting. A constant focus on discharge planning, the lack of time for effective collegiate communication, the setting aside of social determinants, and a perpetual preparedness for the questions of organisational leadership, were behaviours consistent with working in a pressurised 'hospital-demand' environment. Furthermore, the literature attests the participants' experiences of, and responses to, workplace pressure were not unique [30,95], and that these issues are known to have a negative bearing on workforce sustainability and healthcare quality [7]. An implication of this finding is that a more inclusive definition of 'efficiency' might enhance productivity and the quality of outcomes for clinicians, patients, and health systems [8].

Health system decision-makers face the difficult task of balancing a need for meaningful clinician experiences that promote workforce retention and high quality health care, with productivity to ensure financial accountability and sustainability [7,8]. The findings of this research suggest that 'empowerment of self and others' offers a means to rebalance the concept of efficiency and address the imperative of meaningful practice, while also promoting

health system sustainability. Concepts of complexity science, especially its earlier iteration of cybernetics, indicate that clinician autonomy has an important role in optimising efficiency, in the broader sense of this term [73]. The participants had intuitively adopted the value of 'empowerment of self and others', seemingly prompted by the inherent behaviour of 'complexity', that resists constraint and predictability [39,73]. Autonomy seemed to generate a level of 'efficiency' in patient care that 'command and control' leadership could not [73].

Yet, while the notion of 'autonomy' seemingly infers liberty, in this study it implies the "requisite variety" in knowledge, skills, and ideas, needed to work effectively in contexts of ambiguity and complexity [104]. The law of 'requisite variety' is a complexity oriented term developed mid-last century by Robert Ashby, who researched the regulatory behaviour of systems in response to instability [112]. 'The term infers that systems or actors have 'just enough' freedom to generate responses that will match the variety of a complex system or situation [104]. Autonomy at the level of requisite variety fosters adaptive thinking [73], which Briggs and McElhaney [4] deem essential to sound, effective complex care. In turn, effective care, addressing both primary and fundamental patient need, is likely to reduce avoidable hospital representations and patient churn, making the system more 'efficient' and sustainable [83].

Moreover, and this study's findings indicate, positive patient outcomes are associated with positive workplace experiences, which contribute to health workforce attraction and retention [7]. An engaged workforce is more likely to deliver better patient care [77]. These conditions are especially relevant to the complex care workforce, with older, more complex patients accounting for close to half of all hospital admissions [10]. The researchers thus recommend that clinician experience be considered integral to the notion of 'efficiency' in relation to workforce and health system sustainability, and patient outcomes [7,8]. The proposed relationship between clinician experience and system sustainability is illustrated in Fig 2.

In closing this section, it is important to reiterate the need for more research exploring clinicians' experiences of interprofessional, hospital-based, complex care [4]. In addition, with variation in workplace cultures, norms, and team configurations [4], future studies should be considerate of the contextual complexities and nuances shaping clinicians' experiences of caregiving. Using CAP [9] this study has been able to uncover a deep interconnectedness between healthcare system processes, indicators, and norms with clinician experience and practice. While only three participants contributed to this study, the richness of their experiences draws attention to a range of issues that affected their perceptions and practices.

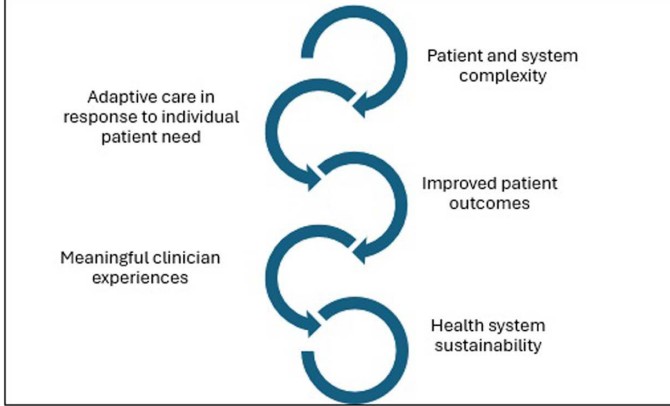

**Fig 2. The complex lifeworld of meaningful practice and health system sustainability.**

These include the importance of giving voice to intellectual diversity in an interprofessional context, and the challenge of doing so. As a result, a potential gap in allied health preparedness was identified, a finding which is supported by other researchers [30,100]. It thus requires educators to consider how allied health professionals might be better informed and empowered for interprofessional practice, since this is increasingly central to health care in hospital settings [24,27,30]. Furthermore, addressing an apparent gap in allied health leadership preparation would also support senior clinicians with the skills to transform emergent clinicians through coaching, education, theoretical knowledge development, and skills building [2]. Such development should aim to build the confidence of clinicians to effect positive change in an environment of uncertainty through the application of evidence-based practice to 'real world' contexts [93]. Normalising the inherent uncertainty in health care may also help to counterbalance the destabilising effects of moral distress [113]. For this reason, the researchers recommend that embedding the expectation of uncertainty in health care should begin at the undergraduate level. Preparing emergent allied health clinicians for uncertainty may also help to overcome junior clinicians' hesitation to engage in clinical discourse, as was witnessed by Jane and Fiona. As Oerlemans et al. [114] advise, moral uncertainty is "an essential phenomenon in medicine", hence, skills to deal with this are equally essential. Finally, this research found that effective complex care entails flexibility and adaptation in clinical practice, while in contrast to the ambulatory study setting, the hospital settings were largely predicated on consistency, predictability, and an overarching hierarchical, albeit congenial order. However, teams that have low levels of hierarchy and bureaucracy and high levels of autonomy tend to be more innovative and adaptive [63]. This research thus challenges notions of efficiency and how these are enacted in different care settings. Perhaps exploring power structures that promote a more horizontal organisational order could deliver greater efficiencies in hospital settings and at the same time, promote better experiences for clinicians and patients. Nevertheless, across each of the study settings, empowerment of self and others was a shared objective among the participants. This highlights how complex systems like perception and experience behave in nonlinear ways that are resistive to constraints, and therefore the need to better understand clinician experience as a profoundly interconnected and influential phenomenon.

## 7. Limitations

This study had several limitations that must be taken into consideration in terms of its findings. First, this was a small, although detailed, study of three allied health professionals' experiences of complex care. The research findings should thus be interpreted on the basis that they are not generalisable in the traditional sense. However, it was not the intention of the researchers to draw generalisable conclusions. Rather, this study aimed to illuminate the generalising effects of social processes [115] within different hospital settings apropos the experiences and practices of the participants. The study's findings should be interpreted in this light.

This study also lacked feedback from the participants on their interview transcripts, which the researchers attribute to the imposts of the coronavirus pandemic. Two of the researchers are healthcare professionals, however, which assisted in the interpretation of participants' narratives. On the other hand, insider knowledge increased the risk of "partial…less reliable, information processing" and researcher bias [Peters in 116]. To address these limitations, the phenomenological techniques of journaling and reflexivity [32] were employed throughout the analytic process. It is nevertheless acknowledged that continued participant input would have enriched this study.

In addition, this study was limited to the experiences of individual clinicians in specific contexts and did not address an extensive array of health system dynamics affecting clinician

experience. These include such challenging issues as hospital 'bed block' [10,30], and the many technology-enabled treatment options rendering clinical decision-making and treatment more complex, costly, time-consuming, and risky [117]. Neither did this research explore the important matter of patients' preferences in, and experiences of, hospital-based interprofessional healthcare [118]. While these issues have a bearing on clinician experience, addressing them all would have exceeded the scope of this study. For this reason, further research into hospital based interprofessional complex care is encouraged. Furthermore, in view of the complexity of this practice, CAP [9] is offered as a suitable framework for such inquiry.

## 8. Conclusion

This research has provided insights into allied health professionals' experience of interprofessional, hospital-based, complex care. The study identified four interconnecting domains of clinician experience that were grounded in patient and systemic complexity: workplace culture, leadership, and empowerment; interprofessional practice; healthcare ethics and values; the ambiguity of complex care. Study findings indicated the value of 'empowerment of self and others' was fundamental to the participants' effective engagement in collaborative practice. Connections were drawn between meaningful practice, patient health outcomes, and notions of healthcare efficiency. It was proposed that investment in allied health leadership development would be an important step for the advancement of interprofessional practitioners. It was also proposed that further research into clinician experience is needed to inform a more holistic interpretation of 'efficiency' in relation to interprofessional, hospital-based, complex care. A more rounded conceptualisation of 'efficiency' may advance efforts toward greater sustainability in patient health outcomes, the health workforce, and the health system.

## Supporting information

**S1 File. Interview question guide.**
(DOCX)

## Author contributions

**Conceptualization:** Felice Borghmans.

**Data curation:** Felice Borghmans.

**Formal analysis:** Felice Borghmans.

**Investigation:** Felice Borghmans.

**Methodology:** Felice Borghmans, Venesser Fernandes.

**Project administration:** Felice Borghmans.

**Resources:** Felice Borghmans.

**Supervision:** Venesser Fernandes, Stella Laletas, Harvey Newnham.

**Writing – original draft:** Felice Borghmans.

**Writing – review & editing:** Felice Borghmans, Venesser Fernandes.

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
