## [Decision Letter · Decision Letter 0]

31 Jul 2024

PONE-D-23-41562"You do need each member of the team to bring that next piece of the puzzle": Allied health professionals' experience of interprofessional complex care in hospital settings.PLOS ONE

Dear Dr. Borghmans,

Thank you for submitting your manuscript to PLOS ONE. After careful consideration, we feel that it has merit but does not fully meet PLOS ONE’s publication criteria as it currently stands. Therefore, we invite you to submit a revised version of the manuscript that addresses the points raised during the review process.

The manuscript has been reviewed by two reviewers and their comments are available below. The reviewers have brought up concerns around the reporting of the manuscript, specifically they have mentioned that the article does not read as a scientific article and this needs to be addressed. Other minor comments around the presentation of the manuscript have also been noted. Please review these comments and make the appropriate revisions. 

We look forward to receiving your revised manuscript.

Kind regards,

Emma Campbell, Ph.D

Staff Editor

PLOS ONE

2. In the online submission form you indicate that your data is not available for proprietary reasons and have provided a contact point for accessing this data. Please note that your current contact point is a co-author on this manuscript. According to our Data Policy, the contact point must not be an author on the manuscript and must be an institutional contact, ideally not an individual. Please revise your data statement to a non-author institutional point of contact, such as a data access or ethics committee, and send this to us via return email. Please also include contact information for the third party organization, and please include the full citation of where the data can be found.

Reviewers' comments:

Reviewer's Responses to Questions

**Comments to the Author**

1. Is the manuscript technically sound, and do the data support the conclusions?

Reviewer #1: Yes

Reviewer #2: Yes

2. Has the statistical analysis been performed appropriately and rigorously? 

Reviewer #1: N/A

Reviewer #2: Yes

3. Have the authors made all data underlying the findings in their manuscript fully available?

Reviewer #1: Yes

Reviewer #2: No

4. Is the manuscript presented in an intelligible fashion and written in standard English?

Reviewer #1: Yes

Reviewer #2: Yes

5. Review Comments to the Author

Reviewer #1: This you for the opportunity to review this study. I think this is such an important topic area as within practice IPE and AHP have had limited focus in previous literature. I look forward to using this publication with my students in the future.

Introduction.

I think the definition of allied health is important as internationally it has different professions within this umbrella term. For example social work is not typically an allied health professional in the united kingdom.

I like the holistic approach to presenting the challenge.

Methods

Table two- I didn’t understand the different colours used in analysis 2 and the significance? Maybe a clearer table with table lines to define continuation from analysis one (if I am picking this up right)

I couldn’t see length of interviews in this section.

I note only 3 participants although given the seniority and diversity in the health service and rich data analysis I respect the contribution to knowledge.

Results

I think the themes interprofessional practice and healthcare ethics (values in sub heading) hit some really hard-hitting issues around ‘vulnerability’ in knowledge gap and listening and respect of patient stories. The section got stronger as it progressed, and the rich quotations added to the section.

Discussion

Figure 1- this Figure was blurry so a better quality is needed. I refrain from Figures in the Discussion and would suggest you consider moving it into results.

I really enjoyed the discussion. It felt robust and I could see person centred thinking reflected within.

Reviewer #2: It looks like the paper submitted is a chapter from a PhD thesis and time was not taken to revise it in a scientific article. Please be so kind to take the time to revise the submission so that it reads like a publication. It is an important piece of work so it does deserve a bit more time and attention. Thank you

6. PLOS authors have the option to publish the peer review history of their article (what does this mean? ). If published, this will include your full peer review and any attached files.

**Do you want your identity to be public for this peer review?** For information about this choice, including consent withdrawal, please see our Privacy Policy .

Reviewer #1: No

Reviewer #2: No

---

## [Author Response · Author response to Decision Letter 1]

27 Nov 2024

In addressing the reviewer’s comments:

The article title has been amended to be consistent with that in the online submission form.

Captions have been included at the end of the manuscript to match figures which have been provided separately as supporting information files. The table of findings has been changed to plain background (the different shades had no particular meaning). A link to deidentified data has been included so that readers may access the data files if they wish. Regarding the suggestion to have figures in the findings section rather than the discussion section, we felt it important to the flow of the document to keep the figures where they were as, from a systems perspective, their intention is to help make sense of how conclusions were drawn.

With respect to the request to make the article more scientific, we have reviewed how we described the study’s conceptual framework and the methods used in data collection and analysis, and framed the discussion accordingly. We hope this meets the editors’ expectations. As this research was qualitative there were no statistical data to present. However, we are happy to make further changes should this be required.

---

## [Decision Letter · Decision Letter 1]

6 Jan 2025

"You do need each member of the team to bring that next piece of the puzzle": Allied health professionals' experience of interprofessional complex care in hospital settings.

PONE-D-23-41562R1

Dear Dr. Borghmans,

We’re pleased to inform you that your manuscript has been judged scientifically suitable for publication and will be formally accepted for publication once it meets all outstanding technical requirements.

Kind regards,

César Leal-Costa, Ph. D

Academic Editor

PLOS ONE

Additional Editor Comments (optional):

Reviewers' comments:

Reviewer's Responses to Questions

**Comments to the Author**

1. If the authors have adequately addressed your comments raised in a previous round of review and you feel that this manuscript is now acceptable for publication, you may indicate that here to bypass the “Comments to the Author” section, enter your conflict of interest statement in the “Confidential to Editor” section, and submit your "Accept" recommendation.

Reviewer #1: All comments have been addressed

2. Is the manuscript technically sound, and do the data support the conclusions?

Reviewer #1: Yes

3. Has the statistical analysis been performed appropriately and rigorously? 

Reviewer #1: N/A

4. Have the authors made all data underlying the findings in their manuscript fully available?

Reviewer #1: Yes

5. Is the manuscript presented in an intelligible fashion and written in standard English?

Reviewer #1: Yes

6. Review Comments to the Author

Reviewer #1: I can see a considerable effort has been made to strengthen this manuscript. It reads much stronger as a result. I still feel figure one is blurry and not a good quality for publication. This is my only recommendation.

7. PLOS authors have the option to publish the peer review history of their article (what does this mean? ). If published, this will include your full peer review and any attached files.

**Do you want your identity to be public for this peer review?** For information about this choice, including consent withdrawal, please see our Privacy Policy .

Reviewer #1: No

---

## [Editor Report · Acceptance letter]

PONE-D-23-41562R1

PLOS ONE

Dear Dr. Borghmans,

I'm pleased to inform you that your manuscript has been deemed suitable for publication in PLOS ONE. Congratulations! Your manuscript is now being handed over to our production team.

Kind regards,

on behalf of

Dr. César Leal-Costa

Academic Editor

PLOS ONE